# A bacteriophage tubulin harnesses dynamic instability to center DNA in infected cells

**Marcella L Erb[1†], James A Kraemer[2†], Joanna K C Coker[1], Vorrapon Chaikeeratisak[1], Poochit Nonejuie[1], David A Agard[2]\*, Joe Pogliano[1]\***

[1]Division of Biological Sciences, University of California, San Diego, La Jolla, United States; [2]Department of Biochemistry and Biophysics, Howard Hughes Medical Institute, University of California, San Francisco, San Francisco, United States

**Abstract** Dynamic instability, polarity, and spatiotemporal organization are hallmarks of the microtubule cytoskeleton that allow formation of complex structures such as the eukaryotic spindle. No similar structure has been identified in prokaryotes. The bacteriophage-encoded tubulin PhuZ is required to position DNA at mid-cell, without which infectivity is compromised. Here, we show that PhuZ filaments, like microtubules, stochastically switch from growing in a distinctly polar manner to catastrophic depolymerization (dynamic instability) both in vitro and in vivo. One end of each PhuZ filament is stably anchored near the cell pole to form a spindle-like array that orients the growing ends toward the phage nucleoid so as to position it near mid-cell. Our results demonstrate how a bacteriophage can harness the properties of a tubulin-like cytoskeleton for efficient propagation. This represents the first identification of a prokaryotic tubulin with the dynamic instability of microtubules and the ability to form a simplified bipolar spindle.

**\*For correspondence:** agard@ msg.ucsf.edu (DAA); jpogliano@ ucsd.edu (JP)

†These authors contributed equally to this work

**Competing interests:** The authors declare that no competing interests exist.

**Reviewing editor**: Jodi Nunnari, University of California, Davis, United States

## Introduction

Tubulins are universally conserved GTPases that polymerize in a head to tail fashion to form filaments. They have evolved properties that allow them to assemble into a wide variety of structures with various functions and dynamics. In eukaryotes, α/β-tubulin forms large, dynamically unstable microtubules (MTs), composed of 13 protofilaments (*Downing and Nogales, 1998*). MTs are the essential component of many cell biological processes; one of the most familiar and well studied is the mitotic spindle required for the faithful segregation of sister chromatids in all eukaryotic cells.

MTs have a distinct growth polarity in which growing (plus) ends elongate substantially faster than minus ends (*Allen and Borisy, 1974*; *Downing and Nogales, 1998*). They are often anchored by their minus ends at specific locations in the cell known as microtubule organizing centers to facilitate mitosis, directional transport, and motility. This anchoring significantly stabilizes the MTs and lends a consistent organization to these structures (*Mitchison and Kirschner, 1984b*). The polarity of the filament becomes key in its ability to specifically interact with binding partners, such as the kinetochore, and for the regulation of the filaments by microtubule accessory proteins (MAPs) (*Euteneuer and McIntosh, 1981*; *Kitamura et al., 2010*). These two features allow for the rapid and effective search and capture of the sister chromatids by the MTs via dynamic instability.

Dynamic instability is defined as the stochastic switching between states of polymerization and rapid depolymerization (*Mitchison and Kirschner, 1984a*). The biomechanical source of this dynamic instability is the differential between the addition of new monomers into the MT lattice and the hydrolysis of the GTP bound to the beta subunit of the heterodimer. While GTP bound to the subunits in the middle of the lattice has been hydrolyzed to GDP, the tip of the growing MT is crowned by a cap of

**eLife digest** For a cell or virus to reproduce, it must duplicate its genome and separate the two copies. In plants and animals, this DNA is stored in the nucleus of each cell in the form of chromosomes, and a complex protein structure called the spindle apparatus is responsible for physically aligning and then separating the duplicated chromosomes.

The spindle apparatus is mainly built from a group of proteins called tubulin, which join together end-to-end to form fibers called microtubules. Proteins that interact with the fibers link them to the chromosomes or to one of the two 'poles' that sit on opposite sides of the cell. Tubulin proteins are then rapidly added to, or removed from, the other end of the microtubule, lengthening or shortening the structure. The rapid switch between fiber growth and shortening is known as dynamic instability. As these fibers shrink, they pull the pair of chromosomes apart and towards opposite sides of the cell.

Viruses that infect bacteria (bacteriophages, or phages for short) replicate their DNA by invading a bacterial cell and hijacking its DNA-replication machinery. In contrast to animals and plants, bacterial cells do not have a cell nucleus, and most bacteria and many phages have genomes comprised of circular DNA rings. Bacteria and phages do share the need to correctly position their DNA in the bacterial cell; but neither bacteria nor phages are known to possess an equivalent to the spindle apparatus that could perform this positioning. A group of bacteriophages were recently found to contain a type of tubulin called PhuZ; however, it was not known if this protein could form a spindle-like apparatus in bacterial cells.

Erb, Kraemer et al. now reveal that PhuZ filaments grow and shorten in similar ways to the microtubules in animal and plant cells: PhuZ proteins are rapidly added to or subtracted from just one end of the fiber. Erb, Kraemer et al. believe this is the first time dynamic instability has been confirmed in a tubulin from a cell without a nucleus.

Erb, Kraemer et al. infected bacteria with bacteriophages that use PhuZ and observed that long fibers of PhuZ form and arrange into a structure that resembles a simple form of the spindle apparatus. These structures only form if a virus has invaded a cell. During the early stages of infection, viral DNA localizes to the cell poles but rearranges at around the same time that PhuZ fibers form. At this point, the viral DNA moves towards the middle of the bacterial cell with fibers on either side of the DNA, suggesting that the fibers play a role in moving the DNA. While the fast growing ends of the fibers were shown to be highly dynamic, the other ends were stably anchored at the cell poles.

Erb, Kraemer et al. propose that this bacteriophage system may be an evolutionary ancestor of the spindle apparatus, although more work is required to confirm this.

GTP-bound heterodimers (*Mitchison and Kirschner, 1984a*). The loss of this GTP cap facilitates a conformational change in the filaments, subsequent depolymerization and, possibly, catastrophe (*Downing and Nogales, 1998*).

Until now, these properties of tubulins have appeared to be unique hallmarks of eukaryotes. Some plasmid encoded bacterial actins, such as ParM and Alp7A, display dynamic instability (*Garner et al., 2004*; *Derman et al., 2009*); however, unlike MTs, ParM filaments elongate bidirectionally (*Garner et al., 2004*) and assemble only transiently during the relatively brief process of pushing plasmid DNA molecules apart (*Campbell and Mullins, 2007*). Furthermore, they are not spatially organized and assemble at random positions in the cell. Several families of bacterial tubulins have been identified that participate in cell division, DNA segregation, and viral DNA positioning (*Larsen et al., 2007*; *Oliva et al., 2012*; *Meier and Goley, 2014*). While TubZ has been shown to treadmill (*Larsen et al., 2007*), for most other prokaryotic tubulins, the type of motion the filament undergoes in vivo remains unclear. None of the bacterial tubulins have been shown to exhibit dynamic instability or other critical properties of a microtubule-based spindle.

We recently identified a family of bacteriophage tubulins, PhuZ, that play a role in spatially organizing DNA during lytic growth and thereby contribute to efficient phage production (*Kraemer et al., 2012*). PhuZ from *Pseudomonas chlororaphis* phage 201ϕ2-1 contains a tubulin fold and an extended C-terminus that forms extensive longitudinal and lateral contacts required to stabilize a unique triple stranded filament (*Kraemer et al., 2012*; *Zehr et al., 2014*). Catalytically defective PhuZ mutants

lacking GTPase activity generate stable filaments and disrupt correct positioning of clusters of DNA during lytic growth (*Kraemer et al., 2012*). While this suggested the importance of filament dynamics, it has remained unclear how PhuZ polymers might facilitate this organization.

Here we present the first example of a prokaryotic tubulin that undergoes dynamic instability both in vitro and in vivo. In addition, the PhuZ cytoskeleton has many of the properties of eukaryotic MTs, including polarity, a GTP cap, and anchoring. These shared properties extend to the ability of both PhuZ and MTs to build a bipolar spindle for the movement of DNA. We further characterize replication of this bacteriophage, demonstrating that the infection nucleoid contains only replicating phage DNA and that nucleoid centering is independent of replication.

## Results

### PhuZ filaments are highly dynamic in vitro

To gain insight into the mechanism of phage centering by PhuZ, we investigated the properties of PhuZ filaments assembled in vitro and in vivo. First, we used total internal reflection (TIRF) microscopy to visualize dynamics of purified PhuZ (*Figure 1A*). In the presence of GTP and a crowding agent to minimize filament diffusion, 2.5 μM PhuZ (20% Cy3-labeled, 80% unlabeled) that was otherwise unattached to PEG-coated coverslips, formed short dynamic filaments that translocated across the coverslip (*Video 1*, *Figure 1A*). Many of these filaments displayed non-uniform intensity, and annealing and severing events were also observed (*Video 2*), implying that many of these structures consist of at least two PhuZ filaments. The apparent motion could be the result of either diffusion near the surface or treadmilling, that is, growth at one end and depolymerization at the other. This ambiguity could be resolved by using the naturally occurring intensity variations along each filament to allow growth and shrinkage rates at each end to be quantified (n = 40) independent of any overall filament motion. This revealed that the filaments treadmill in a coordinated manner (*Figure 1A*), with new filament growth at one end and depolymerization at the other. Depolymerization from these minus-ends occurred at a rate of 15 ± 5 μm/min (n = 40). The large amount of heterogeneity could be due to the presence of bundles and some events being catastrophic depolymerization events from a plus-end. Thus, like MTs and actin, PhuZ filaments must be polar, with one end growing faster than the other. In a manner reminiscent of MT catastrophe, filaments were also observed to occasionally fully depolymerize (*Video 3*).

### PhuZ filaments exhibit dynamic instability and distinct polarity in vitro

To better assess the dynamics and polarity of individual growing filaments, we performed two-color TIRF microscopy of Cy3-labeled PhuZ (green) growing new polymer off of Cy5-labeled and biotinylated seeds (red) in the absence of crowding agent. To make stable seeds, 2 μM PhuZ (20% Cy5, 5% biotin, 75% unlabeled) was polymerized with the non-hydrolyzable GTP analogue GMPCPP for 5 min and attached to PEG-biotin-coated coverslips via streptavidin. After 2 min, the flow chamber was subsequently washed with TIRF buffer (See 'Materials and methods') to remove unattached filaments, free monomers, and GMPCPP. Cy3-labelled PhuZ (20% Cy3, 80% unlabeled) and GTP were then added to the chamber to initiate dynamic filament formation, and dynamics were monitored at 0.25-s intervals for 100 s (*Video 4*, *Figure 1B–D*). To ensure that Cy3-PhuZ filaments would only grow from the preformed Cy5-PhuZ seeds, Cy3-PhuZ was added to the flow chamber at a concentration (1.5 μM) below the critical concentration (2.5 μM). The movies indicate that PhuZ filaments possess a distinct kinetic polarity, with growth only seen from one end, highlighted in *Figure 1*. This is similar to, but more asymmetric than MTs(*Bergen and Borisy, 1980*). The Cy3-PhuZ filaments elongated at a rate of 1.9 ± 0.1 μm/min at 1.5 μM, (n = 40). This rate, when normalized for concentration, is equivalent to the growth rate of ParM filaments(*Garner et al., 2004*), and about 6-fold faster than growing MTs(*Hyman et al., 1992*; *Gupta et al., 2002*; *Bode et al., 2003*) (*Table 1*). After a random period of time, growing PhuZ filaments were observed to switch to rapid disassembly (*Video 4*, *Figure 1C,D*) and completely depolymerize. The remaining stabilized GMPCPP Cy5-PhuZ seed could then nucleate another round of polymerization. As shown in the kymographs (*Figure 1D*), filament depolymerization always (n = 40) went to completion and proceeded at an average rate of 108 ± 20 μm/min. This rate of catastrophic depolymerization is comparable to rates observed for yeast tubulin (103 μm/min)(*Gupta et al., 2002*; *Bode et al., 2003*), but is an order of magnitude faster than that of either mammalian tubulin or ParM (13, 9.4 μm/min, respectively)(*Hyman et al., 1992*; *Garner et al., 2004*). The GMPCPP Cy5-seeds formed stable filaments that never disassembled (*Video 4*, *Figure 1C,D*), showing that, as with MTs,

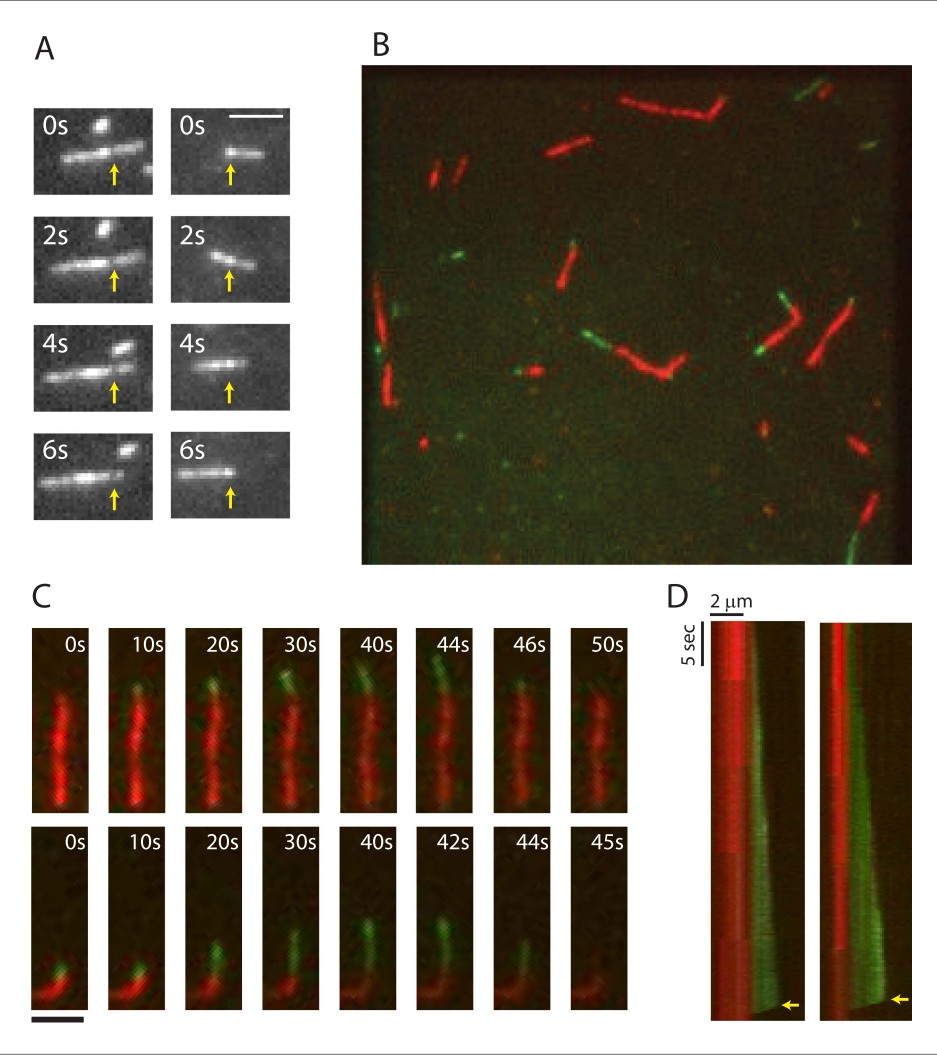

**Figure 1**. TIRF microscopy reveals polarity and dynamic instability in PhuZ filaments. (**A**) Cy3 labeled PhuZ filaments exhibit treadmilling in the presence of GTP. Many filaments displayed non-uniform intensity (highlighted by arrows), which we used as stationary points to monitor growth and shrinkage of the ends independent of filament diffusion. With respect to the highlighted points, one end appears to grow while the other shrinks. (**B–D**) Polymerization of GTP-PhuZ filaments (green) off of GMPCPP stabilized PhuZ seeds (red) (see Methods). (**B**) Wide-field still image of PhuZ filaments growing unidirectionally off of GMPCPP stabilized seeds. (**C**) Montages of two representative PhuZ filaments undergoing dynamic instability. Periods of filament growth are followed by rapid disassembly back to the GMPCPP seed. (**D**) Kymographs of filaments from (**C**). Arrow indicates catastrophe event. Scale bars equal 2 μm.

dynamic instability of PhuZ filaments requires the energy of GTP hydrolysis. PhuZ is the first prokaryotic tubulin known to display dynamic instability.

## Nucleotide hydrolysis drives PhuZ dynamic instability

The dynamic instability observed by TIRF suggests that filament behavior is tightly coupled to nucleotide hydrolysis. Examination of the polymerization kinetics of a catalytically inactive mutant (D190A-PhuZ) by right-angle light scattering (*Figure 2A*) revealed that the critical concentration (320 ± 20 nM) was about 10-fold lower than the 2.5 μM previously measured for wild-type PhuZ (*Kraemer et al., 2012*; *Zehr et al., 2014*). This difference is likely due to the competition between the rate of hydrolysis during transient association of longitudinal dimers (proportional to [PhuZ] (*Allen and Borisy, 1974*)) with the rate of assembly of hexameric nuclei (proportional to [PhuZ] (*Mitchison and Kirschner, 1984a*)) as well as contributions from filament turnover at steady-state due to nucleotide hydrolysis by wild-type

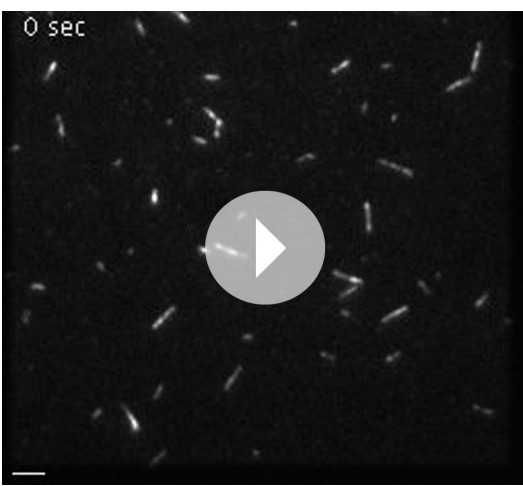

**Video 1**. TIRF microscopy of 2.5 μM Cy3-PhuZ (20% Cy3) filaments reveals PhuZ filaments are dynamic and translocate around the field of view. Filaments treadmill and undergo catastrophic depolymerization. Images were aquired every 250 ms for 75 s. Scale bar equals 2 μm.

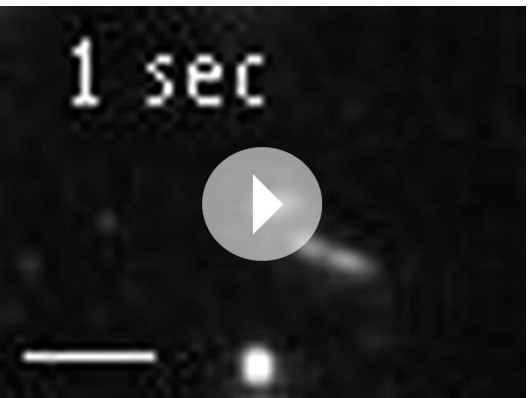

**Video 2**. Close up of Cy3-labelled PhuZ showing annealing, severing, and depolymerization events. Images were acquired every 500 ms for 100 s. Scale bar equals 2 μm.

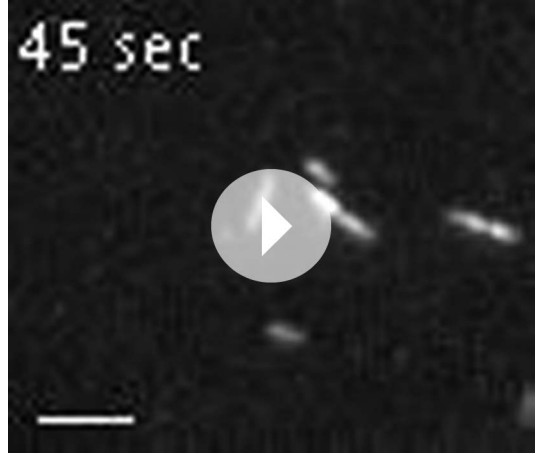

**Video 3**. Close up of Cy-3 labelled PhuZ filaments demonstrating a full depolymerization event. Zoom in from **Video 1**. Scale bar equals 2 μm.

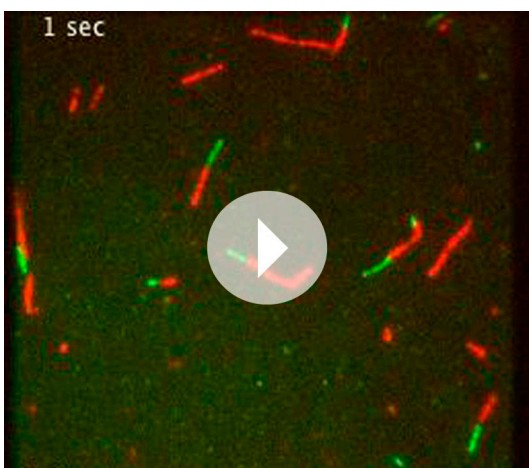

**Video 4**. Dynamic instability and polarity of PhuZ filaments revealed by two-color TIRF microscopy. GMPCPP stabilized PhuZ seeds (20% Cy5, 5% biotin, red) were attached to biotin-PEG coated glass, and 1.5 μM Cy3-PhuZ (20% Cy3, green) and GTP were added. Green filaments are observed to grow from only one end of the seeds and exhibit dynamic instability. Images were acquired every 250 ms for 100 s. Scale bar equals 2 μm.

PhuZ. Unlike wild-type PhuZ, D190A-PhuZ polymerization curves show a linear phase after the initial growth phase (**Figure 2A**), likely due to increased bundling of these non-dynamic filaments. Compared with tubulin, whose GDP critical concentration is about 10x higher than with GTP (**Mitchison and Kirschner, 1984a**; **Caplow et al., 1994**), PhuZ is even more discriminating, with no polymerization detectable with GDP at PhuZ concentrations as high as 200 μM (**Figure 2C**). D190A-PhuZ also requires GTP for polymerization (**Figure 2D**).

While the polymer cannot form with GDP, GTP is rapidly hydrolyzed in the filament body to GDP. To test the stability of the GDP-PhuZ lattice, excess GDP was spiked into wild type or D190A-PhuZ polymerization reactions having 100 μM GTP (**Figure 2B**). Upon addition of GDP, wild-type PhuZ rapidly depolymerized, with no observable polymer detected ~1 min post GDP addition, while no appreciable depolymerization was observed with D190A-PhuZ (**Figure 2B**). Due to the rapid GTP

**Table 1.** In vitro polymerization parameters

| | Pol. Rate (µm/min*µM) | Plus-end Depol. Rate (µm/min) | Minus-end Depol. Rate (µm/min) |
|---|---|---|---|
| PhuZ | 1.9 ± 0.1 (n = 40) | 108 ± 20 (n = 40) | 15 ± 5 (n = 40) |
| Mammalian tubulin | 0.2 (*Hyman et al., 1992*) | 13 (*Walker et al., 1988*) | 7.5 (*Mitchison and Kirschner, 1984a*) |
| Yeast tubulin | 0.18 (*Gupta et al., 2002*; *Bode et al., 2003*) | 103 (*Gupta et al., 2002*; *Bode et al., 2003*) | Not measured |

hydrolysis within the lattice, only the most recently added monomers at the plus-end of the filament contain GTP. Thus, the observed rapid depolymerization after adding GDP implies the existence of a stabilizing GTP-cap that is lost upon GDP addition. The stability observed with D190A-PhuZ demonstrates that the nucleotide state of the penultimate PhuZs must be determining filament stability and not the state of the exchangeable most plus-end monomer. These observations help explain the catastrophic depolymerizations observed in our movies as the result of stochastic loss of the cap due to nucleotide hydrolysis in one or more plus end monomers.

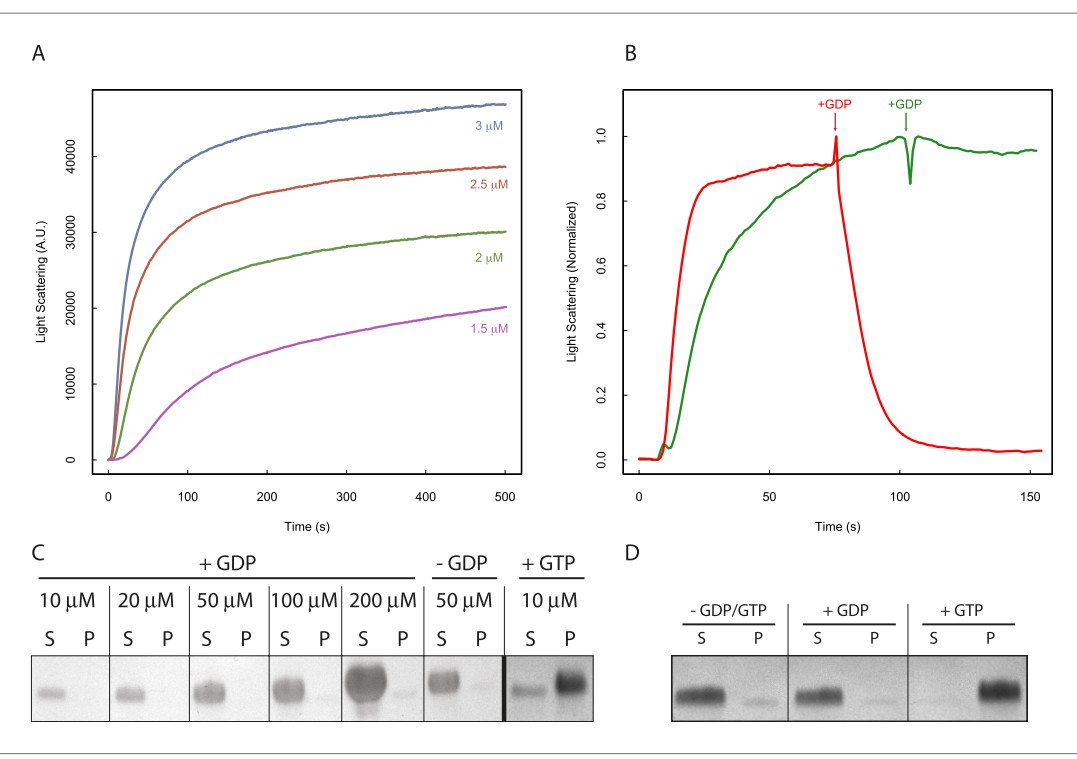

**Figure 2**. Nucleotide hydrolysis destabilizes PhuZ filaments. (**A**) Right-angle light scattering traces of D190A-PhuZ polymerization at 1.5 (purple), 2 (green), 2.5 (red), and 3 (blue) µM upon addition of 1 mM GTP. (**B**) Right-angle light scattering traces of PhuZ (red) or D190A-PhuZ (green) polymerized in 100 µM GTP. 3 mM GDP was added after polymerization reached steady state (arrow + GDP), and subsequent depolymerization was monitored. (**C**) Testing PhuZ polymerization by pelleting (see 'Materials and methods'). Supernatants (S) and pellets (P) were analyzed by SDS-PAGE. Input PhuZ concentration indicated in µM. No detectable PhuZ polymer formed at concentrations as high as 200 µM. No detectable polymer formed in the absence of nucleotide (second from right), and PhuZ filaments were readily detected in the presence of 5 mM GTP (10 µM PhuZ shown right). (**D**) Pelleting of D190A-PhuZ in the presence of 5 mM GDP, 5 mM GTP, or no nucleotide. Nucleotide was added to 10 µM D190A-PhuZ and spun 80000X RPM for 20 min at 25°C. Supernatants (S) and pellets (P) were analyzed by SDS-PAGE. No detectable polymer was formed in the presence of GDP. In contrast, almost all of the protein was found in the pellet in the presence of GTP.

## PhuZ filaments display dynamic instability and are distributed throughout the cell in the absence of phage

To determine if PhuZ displays the microtubule-like properties in vivo that were observed in vitro (unidirectional growth, dynamic instability), we visualized GFP-PhuZ filaments in *Pseudomonas chlororaphis* cells using rapid time-lapse microscopy. To permit fluorescent labeling and visualization, *P. chlororaphis* cells were grown on agarose pads containing arabinose to induce production of GFP tagged PhuZ expressed from a plasmid. When GFP-PhuZ was expressed in the absence of phage infection, but at concentrations comparable to that occurring during an infection, short filaments were formed (average length of 0.9 μm, n = 1260) throughout the cell (*Figure 3A,D*). In time-lapse microscopy, these filaments displayed dynamic instability, undergoing periods of polymerization and depolymerization (*Figure 3B,C*). GFP-PhuZ filaments were not specifically localized (*Figure 3D*) but instead polymerized and depolymerized at positions throughout the cells, as shown by kymographs of fluorescence intensity throughout a 60 s window (*Figure 3C*). In contrast, a catalytically inactive mutant of PhuZ (PhuZD190A (*Kraemer et al., 2012*)) is abolished for filament dynamics (*Figure 3E*). Overall, the dynamic properties of the in vivo assembled filaments of uninfected cells were similar to the results obtained in vitro and filaments displayed no particular organization or strong preference for any region of the cell.

## Upon phage infection, PhuZ filaments become localized and organized into a spindle structure

Upon infection of *P. chlororaphis* cells with phage 201ϕ2-1, PhuZ assembled long filaments that extended approximately half the length of the cell, with an average length of 2.0 μm (n = 780). As

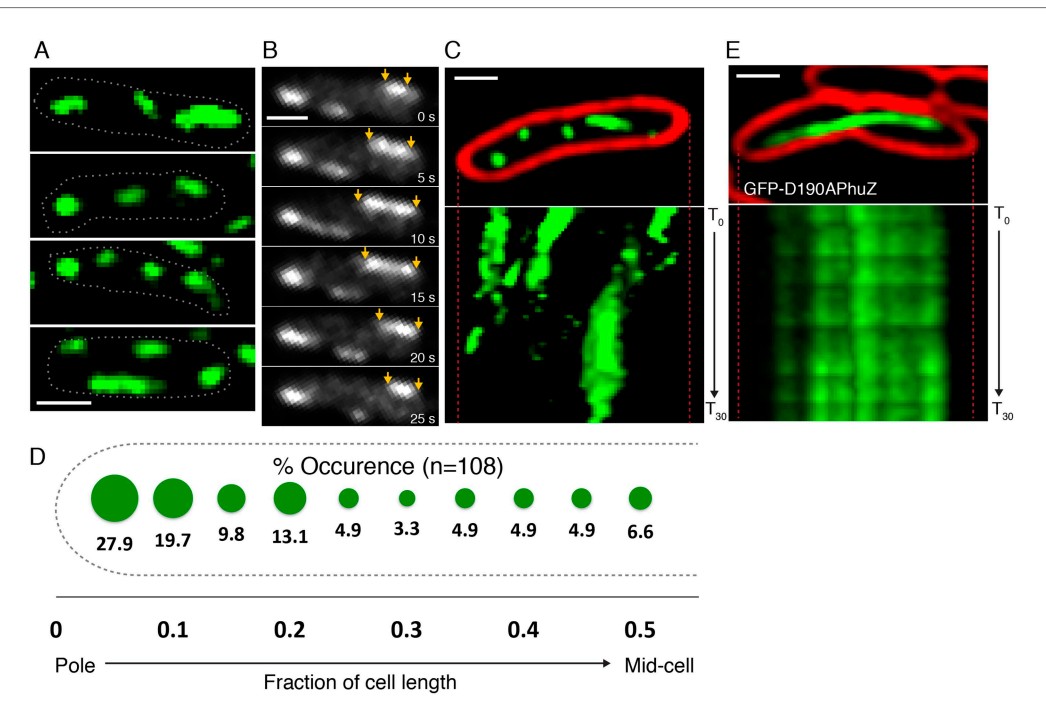

**Figure 3**. PhuZ filaments are distributed throughout the cell in the absence of phage. (**A**) GFP-PhuZ expressed by itself (in the absence of phage infection) assembles relatively short filaments in *P. chlororaphis*. Four individual cells (outlined) with multiple filaments are shown. (**B**) Time-lapse microscopy of a single cell showing an example of a filament (yellows arrows) growing and shrinking over the course of 25 s. (**C**) Kymographs of GFP-PhuZ fluorescence intensity in a single cell throughout a 60 s window. (**D**) Relative positions of filament ends (closest to the cell pole) are expressed as fraction of cell length measured from a field of uninfected cells (n = 110) and plotted as a frequency distribution (fraction of population). (**E**) Kymograph of GFP-D190APhuZ fluorescence intensity in a single cell throughout a 60 s window. The white scale bar equals 1 micron.

shown in *Figure 4A, B*, *Figure 5A* and *Video 5*, PhuZ formed a spindle-like structure in which a pair of bundled filaments emanated from a single location at each cell pole and extended toward the infection nucleoid at midcell. Filaments of the phage spindle were highly dynamic and could be observed to polymerize and depolymerize (*Figures 4B, 5A, 5C, 5E, Video 5*). Dynamics (both polymerization and depolymerization) always occurred on the nucleoid side, with the ends nearest the cell poles appearing relatively static (*Figures 4B and 5A*). While filaments in uninfected cells were located at positions throughout the cell (*Figure 3D*, n = 108), the spindle vertex was almost always (83%, n = 61) stably associated with the extreme pole of the cell (*Figure 4C*). Kymographs of filament position over time in infected cells (*Figure 4B*) suggest that the spindle vertex continuously occupies the cell poles and that the spindle arms reach toward but do not cross the central cavity of the cell occupied by the infection nucleoid. Comparisons of filament length over time in kymographs (*Figures 3C and 4B*) and the average length change per unit time (*Figure 5E*) between uninfected and infected cells suggest a stabilization of the PhuZ filament in the presence of the phage. Although our resolution does not allow us to rule out filament or bundle growth at the vertex of the spindle, one interpretation of these results is that the stabilized minus ends of the filaments are located at the cell pole during phage infection while the dynamically unstable plus ends extend toward the infection nucleoid.

## The infection nucleoid is composed solely of phage DNA and migrates to the center of the Cell

To further understand how PhuZ participates in development and centering of the infection nucleoid and to determine the composition of the DNA in this structure, we followed infection nucleoid formation via fluorescence in situ hybridization (FISH). Using probes complementary to either phage DNA (Cy3) or host chromosomal DNA (Cy5) (*Figure 6A*), we showed that host chromosomal DNA is degraded by

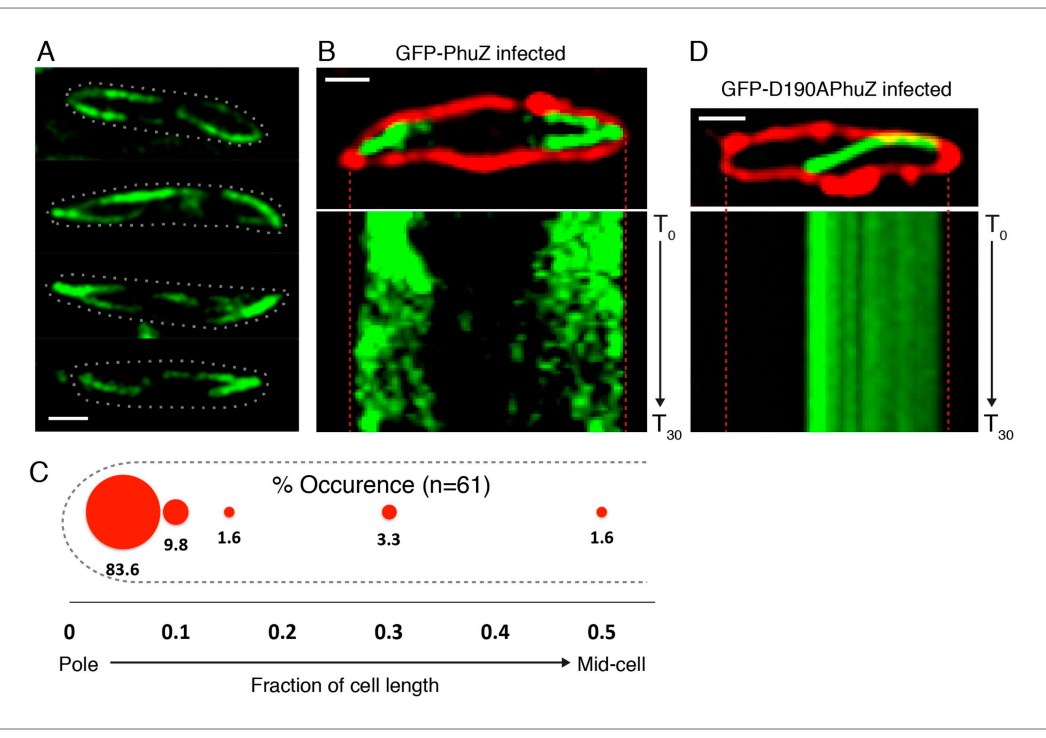

**Figure 4**. PhuZ forms a spindle composed of bipolar filaments in vivo. (**A**) *P. chlororaphis* cells (outlined) that have been infected with phage 201φ2-1 for approximately 60 min have bipolar GFP-PhuZ spindles. (**B**) Kymograph of GFP-PhuZ fluorescence intensity in a single cell infected with phage 201φ2-1 throughout a 60 s window. (**C**) Relative positions of filament ends (closest to the cell pole) are expressed as fraction of cell length measured from several fields of infected cells at a single time point 60 min post infection (red, n = 60) and plotted as a frequency distribution (fraction of population). (**D**) Kymograph of GFP-D190APhuZ fluorescence intensity in a single cell infected with phage 201φ2-1 throughout a 60 s window. The white scale bar equals 1 micron.

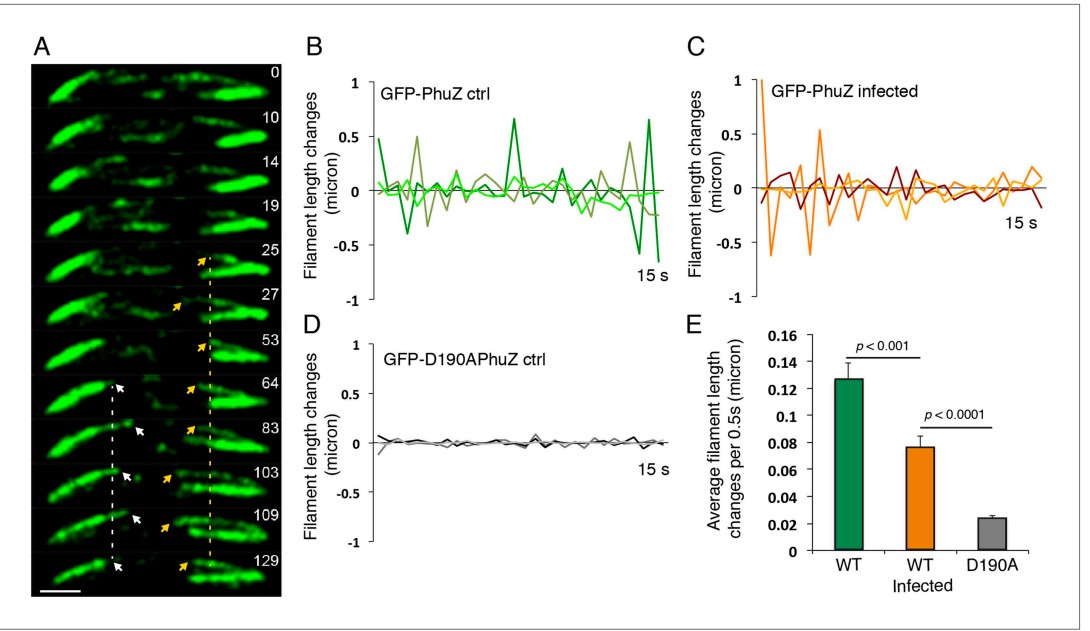

**Figure 5**. Filaments of the PhuZ spindle are dynamically unstable in vivo. (**A**) Time-lapse sequence showing a single cell with a bipolar spindle over the course of 129 s. The filaments of the spindle can be observed to grow and shrink (arrows and *Video 5*). (**B–D**) Length changes of three representative GFP-PhuZ filaments, (**B**) GFP-PhuZ uninfected, (**C**) GFP-PhuZ infected, and (**D**) GFP-D190APhuZ ctrl, throughout 15 s window. The length of the filament was subtracted from the frame before it [n-(n-1)] and plotted over time. (**E**) Average of the absolute value of filament length changes per unit time (0.5 s).

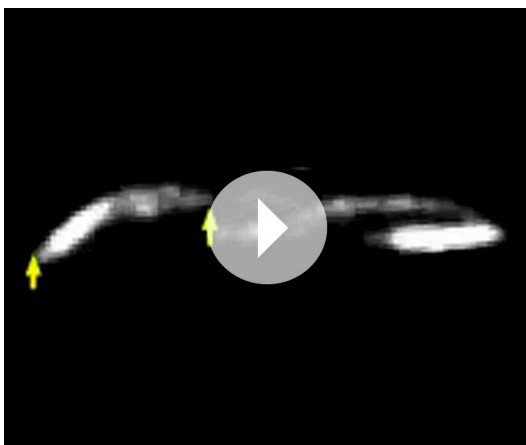

**Video 5**. Time-lapse movie of *P. chlororaphis* cells expressing GFP-PhuZ and infected with phage 201φ2-1 shows that PhuZ filaments form a spindle in which the centrally located ends of the filaments display dynamic instability. Arrows mark the ends of a filament in the left half of the cell. Images were captured 0.5 s apart for 2.5 min. Cells were grown on an agarose pad with 0.15% arabinose to express GFP-PhuZ from the arabinose promoter and then infected with phage.

40 min post infection and that the nucleoid formed during phage infection is composed entirely of phage DNA. We never observed co-localization of the two probe signals. All cells with a phage nucleoid were positive for phage probe binding (*Figure 6B*).

To assess the dynamics of the phage DNA in live cells, we observed infected wild type (WT) cells grown on agarose pads containing the non-specific vital DNA stain Syto16. Upon phage infection, host DNA disappears concomitant with the appearance of a focus of phage DNA, which first appears near one cell pole in 75% of infected cells (n = 155) (*Video 6* and *Video 7*, *Figure 7A*). Over time, the small circular mass of phage DNA increases in size and migrates toward the cell midpoint (*Figure 7A,B*). After arrival at midcell, the phage nucleoid oscillates slightly about the central axis (*Figure 7C,D*), suggesting that the nucleoid continues to experience positioning forces. This oscillatory movement is reminiscent of eukaryotic chromosomes lined up at midcell by the mitotic spindle. In contrast to eukaryotes, where DNA replication is temporally separated from segregation, the phage nucleoid continued to increase in mass as it moved toward midcell, suggesting that DNA replication and positioning occur simultaneously.

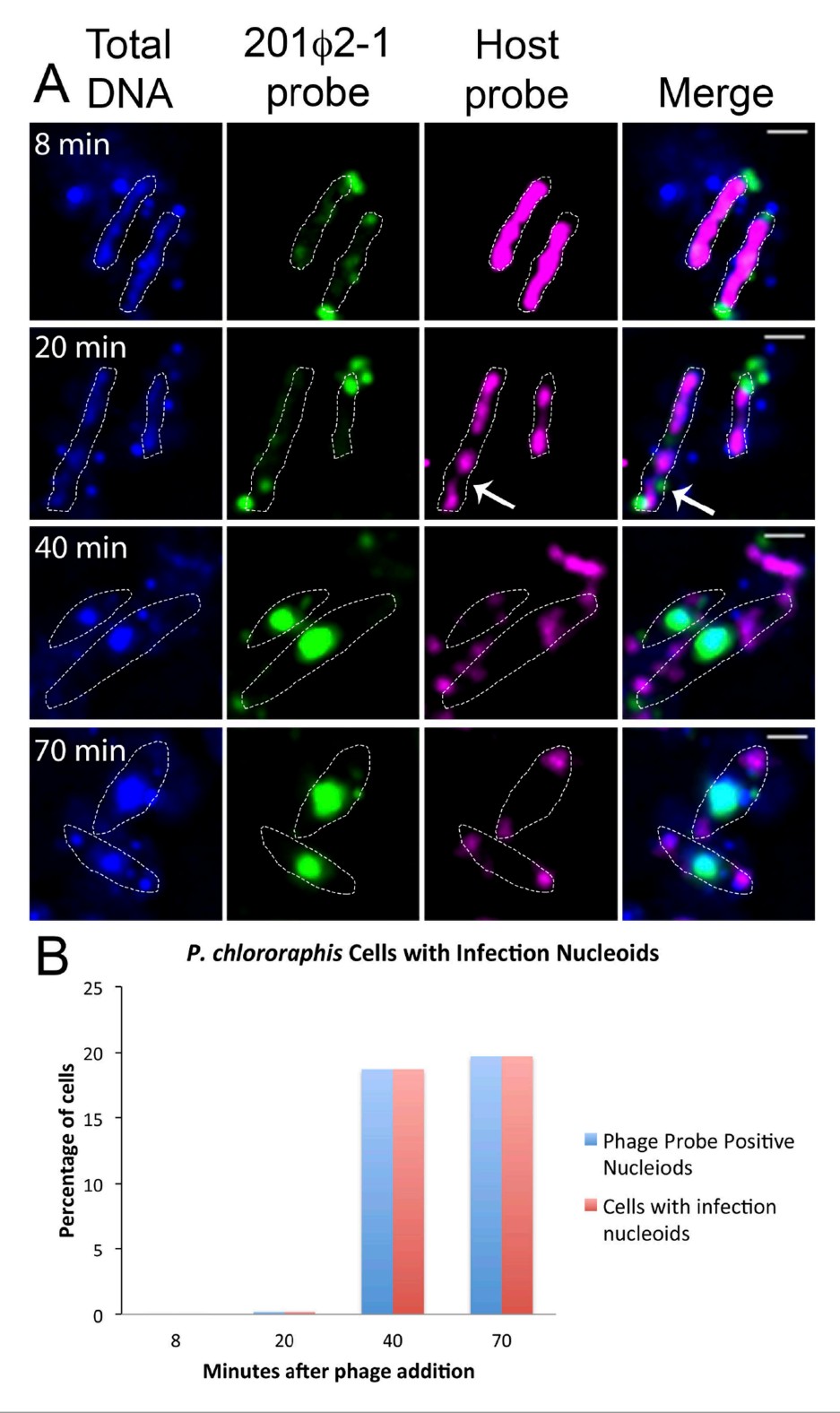

**Figure 6**. Fluorescence in situ hybridization (FISH) during phage infection. (**A**) Infected *P. chlororaphis* cells were fixed and hybridized with DNA probes specific for either the host chromosomal DNA (pink, labeled with Cy3) or phage 201φ2-1 DNA (green, labeled with Cy3). Total DNA was stained with DAPI, shown in the first column. Scale bar equals 1 micron. Cells outlines are indicated with a white dotted line. By 40 min post infection, host DNA was

*Figure 6. Continued on next page*

*Figure 6. Continued*

mostly degraded and only small remnants of *P. chlororaphis* DNA was detectable, typically near the cell poles. (**B**) Graph describing development of infection nucleoids over time during infection. All infection nucleoids stained with phage probe.

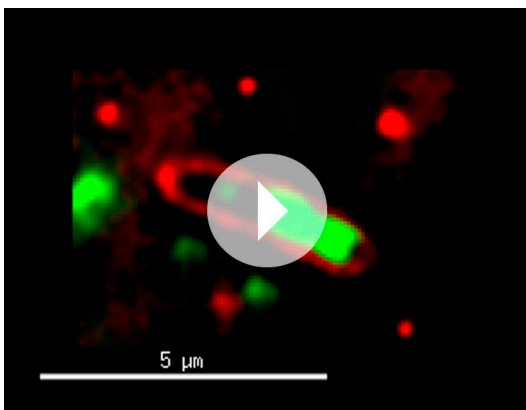

**Video 6**. Time-lapse movie corresponding to the cell in *Figure 4A* showing development of a phage nucleoid in a single infected cell over the course of 63 min. The phage nucleoid first appears as a small green focus at the cell pole that migrates to the cell midpoint and develops into a very large infection nucleoid. Images were captured 7 min apart for 63 min. Cells were grown on an agarose pad containing Syto16 and then infected with phage. This example of nucleoid formation corresponds to the nucleoid shown in *Figure 7A*.

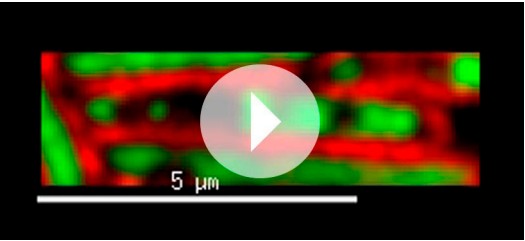

**Video 7**. Time-lapse movie showing development of a phage nucleoid in a single infected cell over the course of 49 min. This example of nucleoid formation and movement corresponds to one of the nucleoids reported in the graph in *Figure 7C*. The phage nucleoid first appears at the cell pole and then migrates to cell midpoint. Images were captured 7 min apart for 49 min. Cells were grown on an agarose pad containing Syto16 and then infected with phage.

## Phage nucleoid positioning is independent of DNA replication

The infection nucleoid is composed solely of phage DNA (*Figure 6*), increases in size (replicates) as it moves toward midcell (*Figure 7*) and is likely the site of genome encapsidation (*Kraemer et al., 2012*). We therefore sought to test whether replication was required, directly or indirectly, for phage DNA movement. At various points during infection we added a potent DNA gyrase inhibitor (ciprofloxacin) that blocks replication of DNA in a broad spectrum of bacteria, plasmids and phage (*Alonso et al., 1981*; *Constantinou et al., 1986*; *Fisher et al., 1989*). While ciprofloxacin reduced phage nucleoid size as expected, indicating a reduced accumulation of DNA (*Figure 8A*), it had no effect on its positioning at midcell, suggesting that DNA replication was not required for the centering function of the PhuZ spindle (*Figure 8B*). By contrast, interfering with PhuZ filament dynamics has a marked affect on phage DNA positioning (*Kraemer et al., 2012*).

## Discussion

The ability of intrinsically polarized and dynamically unstable MTs to be anchored, stabilized, and regulated allows these filaments to be harnessed by the eukaryotic cell to perform key organizational tasks. It has been argued that the spatiotemporal organization of the cytoskeleton by organizing centers may be a defining characteristic of eukaryotes (*Theriot, 2013*). However, the prokaryotic cell also utilizes cytoskeletal proteins to organize internal space and efficiently execute essential life processes. Now we demonstrate that even entities often considered 'non-living', such as bacteriophage, can exploit the advantages of a well-defined cytoskeletal organization to faithfully propagate themselves.

We have shown that the tubulin PhuZ polymerizes into a filament that is intrinsically polar and dynamically unstable and that in vivo it is anchored and stabilized at the cell poles. Furthermore, it assembles into a bipolar spindle that more than superficially resembles its eukaryotic counterpart. The PhuZ spindle appears to play a key role in organizing viral reproduction by positioning the phage nucleoid at midcell (*Figure 9*). Since polar localization is phage infection-dependent, we hypothesize that a phage protein (depicted as purple triangles in *Figure 9*) is necessary for the organization of the spindle. Based on our long time lapse movies of complete infection cycles, a transition must occur between early un-localized pre-existing

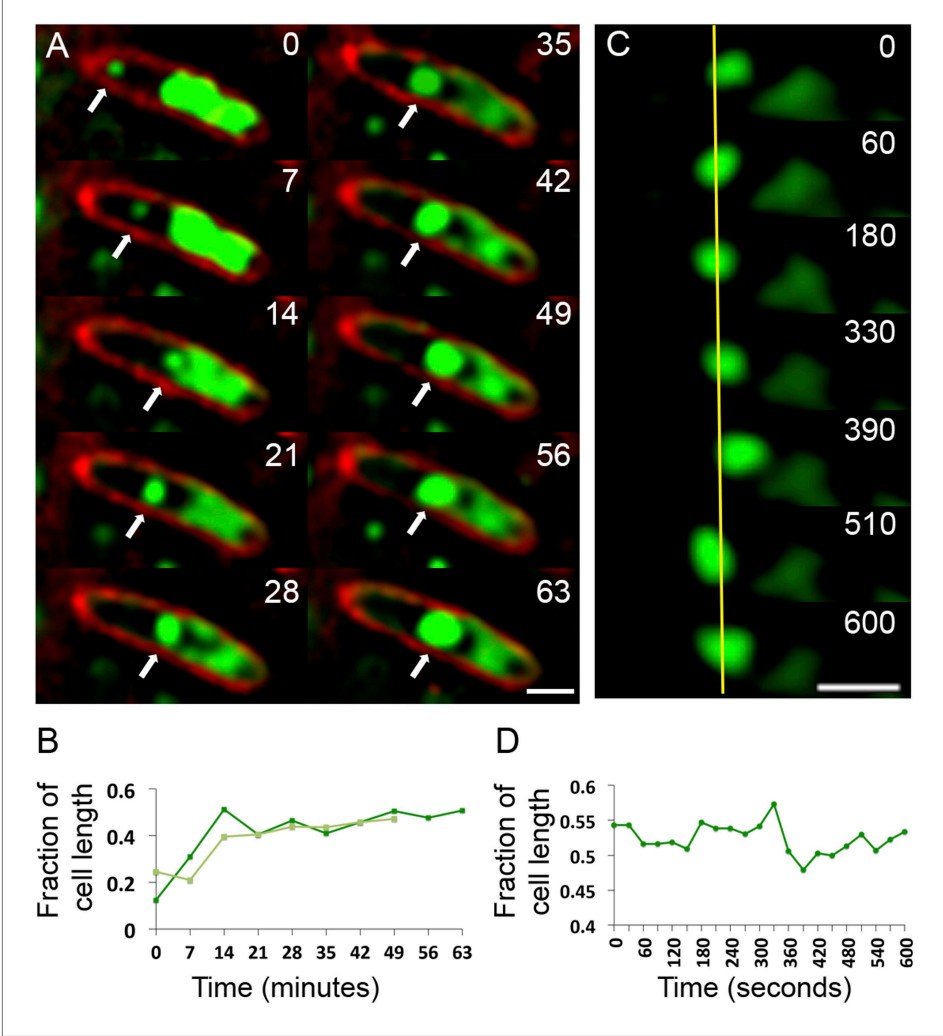

**Figure 7**. Observation of phage nucleoid migration by time-lapse microscopy. (**A**) Time-lapse sequence showing development of a phage nucleoid (arrows) in a single cell over the course of 63 min. Membranes are stained red with FM 4-64 and the DNA is stained green with Syto16. At time zero, a small phage nucleoid (small green focus, arrow) is observed near the cell pole in *P. chlororaphis* infected with phage 201ϕ2-1. Over time the phage nucleoid moves to midcell while it increases in size. The host chromosome, which fills half of the cell at time zero (large green mass), is degraded during infection (*Video 6*). The scale bar equals 1 micron. (**B**) Position of the phage nucleoid in panel A (dark green) is recorded as fraction of cell length and plotted vs time. A second example of nucleoid migration is plotted in light green and is shown in *Video 7*. (**C**) Time-lapse showing a representative example of nucleoid oscillation in a late stage infected cell over the course of 600 s. (**D**) Position (expressed as fraction of cell length) of the phage nucleoid in panel C plotted vs time shows nucleoid movement. Scale bars equal 1 micron.

filaments to the localized polar structures seen in late infection, suggesting a clear role for anchoring. Whether the putative phage polar protein also acts to nucleate de novo filament formation or facilitate bundling is unclear. We propose that early during infection, the dynamically unstable ends of PhuZ polymers (plus ends, by analogy with tubulin) specifically interact with and move the replicating phage DNA, presumably by applying pushing forces, although this movement is independent of replication.

Tubulins in bacteria have been studied intensively for nearly 25 years. To date, prokaryotic tubulins have either been demonstrated to treadmill (*Larsen et al., 2007*) (TubZ) or the relevant filament movement remains an unresolved matter (FtsZ). This apparent switch in the type of functional movement displayed by prokaryotic tubulins versus eukaryotic ones had posed an intriguing question—why don't bacterial tubulins undergo dynamic instability? We now know that some of them do and this particular

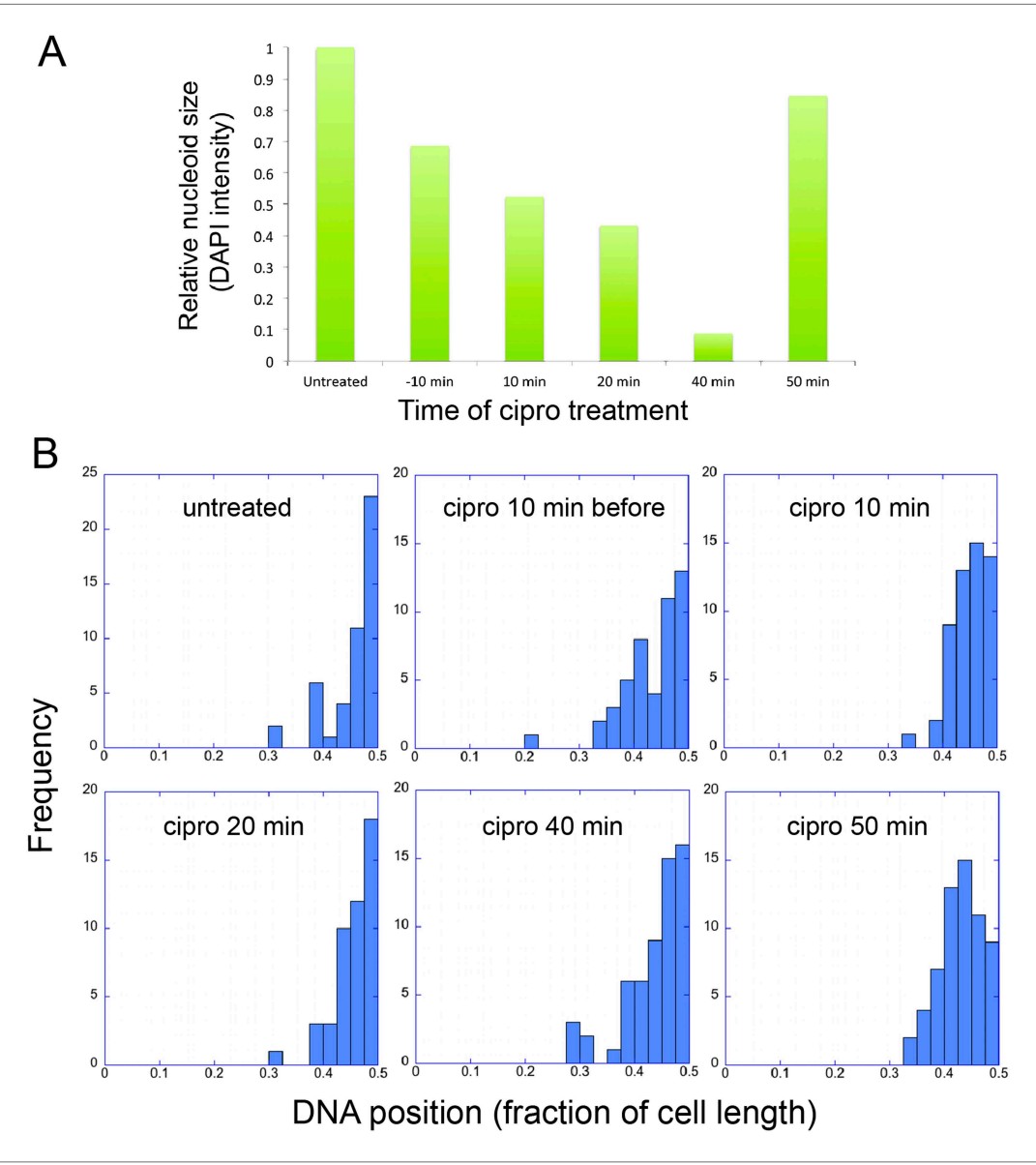

**Figure 8**. Infection Nucleoid Centering is Independent of DNA replication. (**A**) DNA content of infection nucleoids was measured by DAPI staining. Cells were grown on an agarose pad and infected with phage 201φ2-1. Ciprofloxacin was applied to cells 10 min prior to infection or at various times (10, 20, 40, and 50 min) after infection. After 80 min of infection, cells were fixed and images were collected. Total DAPI intensity was measured for approximately 300 cells for each time point. The average total DAPI intensity was normalized to the value for untreated cells. The addition of ciprofloxacin inhibited phage nucleoid (DAPI intensity) growth when added during the first 40 of minutes phage infection, but had little effect on replication when added at 50 min, suggesting that by 50 min DNA replication was mostly completed or no longer dependent on DNA gyrase. (**B**) Histograms showing the position of phage DNA within the cell plotted as a fraction of cell length vs the percentage of the population (frequency) for each time point in (**A**). Although ciprofloxacin treatment inhibited phage DNA replication, it had no effect on phage DNA positioning.

type of biomechanics may inherently lend itself to the formation of complex structures for the search and capture of large masses of DNA.

PhuZ appears to be unique from other prokaryotic systems implicated in DNA positioning in prokaryotes. The tubulin based plasmid segregation system TubZ has not been shown to display dynamic instability (*Larsen et al., 2007*; *Chen and Erickson, 2008*). TubZ assembles two and four-stranded

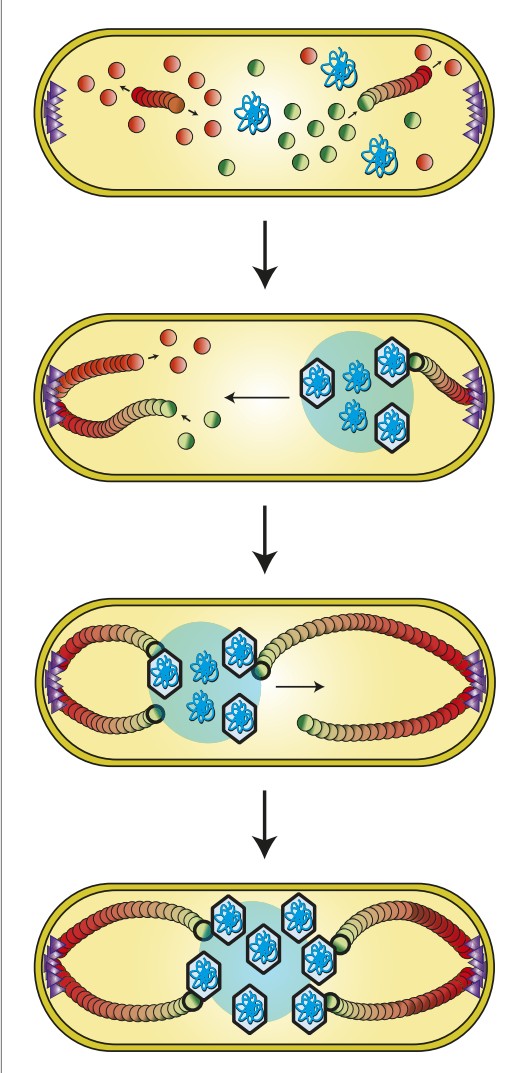

**Figure 9**. Model of PhuZ bipolar spindle formation during lytic growth. PhuZ is expressed early in lytic growth and forms dynamically unstable polymers anchored at the cell poles. Filaments polymerize unidirectionally, with GTP-bound (green) subunits adding to one end, to center the replicating phage DNA at midcell. Capsids assemble in the phage nucleoid for DNA packaging. We speculate that a yet to be identified organizing factor (purple triangles) is required for spindle assembly.

polymers (*Montabana and Agard, 2014*) in contrast to the dynamically unstable triple stranded filaments of PhuZ (*Zehr et al., 2014*). Several actin based plasmid segregation systems display dynamic instability in vitro and in vivo, but unlike PhuZ, these proteins only transiently assemble spindles that rapidly push copies of their plasmids to the cell poles (*Moller-Jensen et al., 2002*, *2003*; *Garner et al., 2004*; *Campbell and Mullins, 2007*; *Derman et al., 2009*; *Sengupta et al., 2010*). Proteins belonging to the highly conserved ParA ATPase family segregate plasmids, chromosomes, and protein complexes in a variety of organisms (*Gerdes et al., 2010*; *Schumacher, 2012*; *Kiekebusch and Thanbichler, 2014*). ParA proteins have been proposed to form ATP dependent concentration gradients or other intracellular structures as they carry out these diverse functions, but none of them form dynamically unstable polymers similar to PhuZ (*Lim et al., 2014*; *Vecchiarelli et al., 2014*). Clearly, bacteria and their mobile genetic elements have evolved an enormous number of nucleotide hydrolyzing, self-assembling, DNA positioning machines. This raises the question as to whether any eukaryotic cells have inherited one of these many ancient DNA positioning mechanisms.

All eukaryotic cells are thought to utilize a mitotic spindle to ensure accurate inheritance of sister chromatids to the daughter cells, but it has long been a mystery how such a complicated structure might have evolved. The first function of the eukaryotic spindle is to line up replicated chromatids at the midline of the dividing cell. Our discovery of a bacteriophage that assembles a similar structure raises the possibility that bipolar spindles arose more than once by convergent evolution taking advantage of the unique properties of tubulin polymers. Alternatively, it is also possible that phage hijacked the mitotic spindle from an ancient eukaryotic cell. However, since bacteria and phage are thought to predate eukaryotic life, an intriguing possibility is that the mitotic spindle first evolved in a bacteriophage for the purpose of positioning viral DNA during lytic growth and was later co-opted for positioning chromosomal DNA in the progenitor of the first

eukaryotic cell. One can imagine that a primordial tubulin evolved to form filaments of increasing biochemical, structural, and functional complexity, beginning with single stranded protofilaments (FtsZ) required for cell division (*Erickson et al., 1996*; *Meier and Goley, 2014*). These then evolved into multi-stranded filaments capable of segregating plasmid DNA (TubZ) (*Larsen et al., 2007*; *Chen and Erickson, 2008*; *Aylett et al., 2010*; *Oliva et al., 2012*; *Montabana and Agard, 2014*) and centering viral DNA (PhuZ). The increased structural and biochemical complexity, including an increase in the number of protofilaments, would allow the lattice to become more cooperative and provide a greater difference between growth and nucleation rates. This in turn would allow the lattice to become more metastable, thereby storing the greater energy necessary for more complex functions within the lattice.

Bacterial tubulin homologs have proven as essential to prokaryotic cell biology as to eukaryotes, and as such, it is not totally unexpected that the advantages conferred upon a cell or virus by the ability to build complex tubulin based structures would also be selected for and shared across kingdoms.

## Materials and Methods

### Protein expression and Purification

Wild-type and KCK-PhuZ were expressed and purified as previously reported (*Kraemer et al., 2012*). D190A-PhuZ was purified by an altered protocol to minimize polymerization. Cultures were lysed in a buffer containing 500 mM KCl, 2 mM EDTA, 10% glycerol, 15 mM thioglycerol, and 50 mM HEPES, pH 8 and cleared at 35k xg. 0.1 vol of DOWEX resin were then added to the clarified lysate to remove nucleotide. The sample was then spun at 38,000 RPM in a Ti45 rotor (Beckman Coulter, Pasadena, CA) to remove the resin and residual aggregates. D190A-PhuZ was then purified by Ni-affinity chromatography using an EDTA-resistant Ni-resin (Roche, Switzerland). The 6× His-tag was cleaved with thrombin protease and EDTA was dialyzed out, and the protein was subsequently purified by gel filtration (Superdex 200) in a buffer containing 250 mM KCl, 1 mM MgCl$_2$, 10% glycerol, 15 mM thioglycerol, and 50 mM HEPES, pH 8. Prior to experiments, all constructs were spun at 80,000 RPM in a TLA100 rotor (Beckman) at 4°C for 20 min.

### Dye and biotin labeling of PhuZ

Thioglycerol was removed from KCK-tagged protein by a Zeba column (Pierce) equilibrated with buffer with no reducing agent. A 2-fold molar excess of dye, Cy3- or Cy5-maleimide (GE, Fairfield, CT), or biotin-maleimide (Sigma-Aldrich, St. Louis, MO) was added and incubated for 15 min at 25°C. 50 mM DTT was added to quench the reaction and the reaction was spun at 80,000 RPM in a TLA100 rotor (Beckman) to remove aggregates. To remove excess dye and non-functional protein, labeled PhuZ was exchanged into BRB80 pH 7.2 in by a Zeba column (Pierce, Rockford, IL), polymerized by the addition of 5 mM GTP, and pelleted at 80,000 RPM in a TLA100 rotor (Beckman). Non-polymerized protein was removed, and the pellet was resuspended in a depolymerization buffer (500 mM KCl, 1 mM MgCl$_2$, 15 mM thioglycerol, 10% glycerol, 50 mM HEPES pH 8) on ice for 1.5 hr. Labeled protein was buffer exchanged into BRB80 pH 7.2 and stored.

### Preparation of PEG-coated Glass slides

Slides were prepared using a modified protocol from *Bieling et al. (2007)*. 24 × 40 mm coverslips (VWR, Radnor, PA) were sonicated in 2 M KOH for 30 min. Slides were then washed three times with H$_2$O and sonicated in piranha solution (2 parts 30% H$_2$O$_2$, 3 parts H$_2$SO$_4$) for 30 min. Slides were subsequently washed three times with H$_2$O and spun dry. Slides were silanized by making sandwiches with a drop of GOPTS (Sigma) and baked for 1 hr at 75°C. Sandwiches were separated and washed in dry isopropanol (Sigma), and spun dry. 30 µl of a saturated PEG-SEV acetone solution (1% biotin-PEG-SEV) (Laysan, Arab, AL) were added to silanized slides, and sandwiches were baked for 4 hr at 75°C. Sandwiches were separated in H$_2$O, sonicated for 5 min, spun dry, and stored in the dark.

### Total internal reflection fluorescence (TIRF) microscopy

Flow chambers were made using double-sided tape. For single color experiments, the chamber was washed three times with imaging buffer (BRB80 pH 7.2 supplemented with 100 mM KCl, 0.5% BSA, 0.5% methylcellulose, and 40 mM βme). 4 mM GTP was added to 2.5 µM Cy3-PhuZ (80% wild-type, 20% Cy3-labeled) in imaging buffer, flowed into the chamber, and subsequently imaged. Images were acquired at a 0.5 s interval on an Andor CCD camera.

For two-color imaging, chambers were washed three times with imaging buffer (supplemented with a GLOX system to minimize photobleaching), followed by 5 min incubation with neutravadin. Chambers were then washed three more times with imaging buffer to remove unbound neutravadin. 200 µM GMPCPP was added to 2 µM Cy5-PhuZ (75% wild-type, 20% Cy5, 5% biotin) to induce polymerization. After 5 min on ice, Cy5-PhuZ-GMPCPP seeds were added to the chamber. After 2 min, the chamber was washed three times with imaging buffer to remove any seeds not adhered to the cover slip. 4 mM GTP was added to 1.5 µM Cy3-PhuZ (80% wild-type, 20% Cy3), added to the chamber, and imaged at a 0.25 s interval.

## Light scattering

Right angle light scattering was conducted by mixing D190A-PhuZ with BRB80 pH 7.2 containing GTP using a stop-flow system designed in-house. An excitation wavelength of 530 nm was used. The critical concentration was determined by plotting the maximum intensity vs PhuZ concentration. The x-intercept of this plot was used as the critical concentration.

For nucleotide spiking experiments, protein and buffer were mixed 1:1 (150 µl reactions) by hand and polymerization was followed by right angle light scattering. Upon reaching plateau, 10 µl of GDP or buffer was added.

## Pelleting

5 mM GDP or GTP was added to PhuZ protein and, after 5 min, samples were spun at 80,000 RPM in a TLA100 rotor (Beckman) for 20 min at 25°C. Supernatants were carefully removed and the pellet was resuspended in 1× gel loading buffer. Samples from the supernatant and pellet were then run on a 10% SDS-PAGE gel and stained with Simple Blue (Life Technologies).

## Live cell microscopy

*Pseudomonas chlororaphis* cells were grown on 1% agarose pads supplemented with 25% Luria broth, 1 µg/ml FM4-64, and when appropriate, either 0.5 µM Syto 16 or 5 µg/ml DAPI as described (*Kraemer et al., 2012*). For phage infections, 5 µl of phage lysate ($10^8$ pfu/ml) was applied to the cells, a coverslip was added, and images were captured using a DeltaVision Spectris Deconvolution microscope (Applied Precision, Issaquah, WA). GFP-PhuZ was expressed in strain ME41 (*Kraemer et al., 2012*) from the arabinose promoter by including arabinose within the pad at a concentration ranging from 0 to 2%, as indicated.

## Fluorescence in situ hybridization (FISH)

*P. chlororaphis* cells were grown on a 1% agarose pad, infected with 5 µl of phage 201φ2-1 lysate ($10^8$ pfu/ml) and at various times (8, 20, 40 70 min) after infection fixed with glutaraldehyde (0.025%) and paraformaldehyde (16%). Fixed cells were processed for hybridization as described (*Ho et al., 2002*). Briefly, cells were washed in PBS, blocked at 75°C for 2 min with 70% formamide, 2× SSC, 1 mg/ml Salmon sperm DNA, washed one time each with 70%, 90%, and 100% ethanol, and allowed to dry. Cells were then treated with 50% formamide, 2× SSC for 5 min at 23°C, probe was added, and then heated at 94°C for 2 min, and then hybridized at 42°C overnight. DNA probes specific for the host *P. chlororaphis* chromosomal DNA or phage 201φ2-1 DNA were prepared by first digesting total *P. chlororaphis* chromosomal DNA or total phage 201φ2-1 DNA with a set of enzymes (*Bsp1286 I, HhaI, HpyCH4 III, Hpy188 I, Nla III*), and then labeling the ends with Cy3-dCTP or Cy5-dCTP using terminal deoxynucleotidyl transferase.

## DNA replication inhibition by ciprofloxacin

*Pseudomonas chlororaphis* cells were grown on 1% agarose pads supplemented with 25% Luria broth, 1 µg/ml FM4-64, and 5 µg/ml DAPI. After 2 hr of cell growth at 30°C, 5 µl of phage lysate ($10^8$ pfu/ml) was applied to the cells, and infection was allowed to proceed for 80 min at 30°C. 5 µl of ciprofloxacin (2 µg/ml) was added 10 min before infection, or 10, 20, 40, and 50 min after infection. After 80 min of infection, cells were fixed on the pad with glutaraldehyde (0.025%) and paraformaldehyde (16%) and then coverslips were added and the cells imaged using a DeltaVision Spectris Deconvolution microscope (Applied Precision, Issaquah, WA). As a control, a set of infections were performed in which only buffer (1 N HCl) but no ciprofloxacin was added. In addition, we also examined cells treated with ciprofloxacin but without the addition of phage. Images were analyzed using ImageJ software to quantitate DAPI intensity of the infection nucleoids and to determine the position of infection nucleoids relative to cell lengths.

## Acknowledgements

We thank members of the Agard and Pogliano labs for invaluable discussions, especially Daniel Elnatan for TIRF microscopy assistance. We also thank members of the Mullins lab for TIRF advice. The OMX and Light Microscopy Facility at UCSD is funded by grant NS047101. We thank Jennifer Santini for her invaluable help with the OMX. JAK. was supported by the Genentech Foundation Predoctoral Fellowship and the Achievement Rewards for College Scientists Foundation Award. This work was

supported by the Howard Hughes Medical Institute (DAA.) and National Institutes of Health grants R01GM073898 (JP), GM031627 (DAA), and GM104556 (JP and DAA).

## Additional information

### Funding

| Funder | Grant reference number | Author |
|---|---|---|
| National Institute of General Medical Sciences | R01GM073898 | Joe Pogliano |
| Howard Hughes Medical Institute | | David A Agard |
| National Institute of General Medical Sciences | GM031627 | David A Agard |
| National Institute of General Medical Sciences | GM104556 | David A Agard, Joe Pogliano |

The funders had no role in study design, data collection and interpretation, or the decision to submit the work for publication.

### Author contributions

MLE, JAK, Conception and design, Acquisition of data, Analysis and interpretation of data, Drafting or revising the article; JKCC, VC, Conception and design, Acquisition of data, Analysis and interpretation of data; PN, DAA, JP, Conception and design, Analysis and interpretation of data, Drafting or revising the article

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
