## [Decision Letter]

Thank you for sending your work entitled “A Bacteriophage Tubulin Harnesses Dynamic Instability to Center DNA in Infected Cells” for consideration at *eLife*. Your article has been evaluated by Richard Losick (Senior editor), a Reviewing editor, and 2 reviewers.

As you will see from the appended reviewers' comments, there was a high level of interest in your study describing how a bacteriophage tubulin creates spindle-like structures in bacteria to position phage DNA. However, the reviewers felt that key data fell short of supporting your model and thus revisions are required before the manuscript is suitable for publication. Specifically, conclusions regarding dynamic instability and organization of filaments in spindle-like structures of infected cells made on the basis filament tracking data require substantiation. Reviewer 2 makes specific suggestions of alternative image analysis strategies that could address these concerns. Both reviewers also made other excellent suggestions that will improve the quality/clarity of the manuscript that should be considered.

Reviewer #1:

This interesting paper from Erb, Kraemer et al reports the dynamic properties of a tubulin isotype, PhuZ, encoded by the genome of a bacteriophage that infects Pseudomonas chlororaphis bacteria. The authors report that filaments of PhuZ, like eukaryotic MTs, display dynamic instability *in vivo* and *in vitro*, and that within the bacterial cell during lytic infection, they assemble into a spindle-like structure capable of using free energy released by PhuZ polymerization to generate polar-ejection forces that push the replicating phage DNA to the center of the bacterial cell. I think this is a well-written, succinct, scientifically sound and interesting study. My only minor concern is the extent to which it advances what was published in their Cell paper, but on balance I think it does represent a sufficiently significant advance, and accordingly I recommend publication in *eLife*. I have only minor suggestions for the authors to consider:

1) It would be useful to present a small summary table comparing the four parameters of dynamic instability (i.e. growth rate, shrinkage rate, catastrophe frequency and rescue frequency) displayed by the phage tubulin compared to eukaryotic tubulins.

2) Figure 7 shows a drawing that depicts some type of PhuZ organizing center that anchors the minus ends at the poles of the cell. What is the evidence for this, and are there any molecular candidates for this function (in the Shapiro lab model for the ParA spindle, TipN fulfills this function)?

3) I think the discussion would be more interesting if the authors noted that other workers have also proposed the existence of spindle-like structures that are responsible for prokaryotic DNA segregation, e.g. by using actin-like filament depolymerase motors to mediate plasmid segregation e.g. Ptacin et al, 2010, Nat Cell Biol. 12:791. Moreover, other workers have proposed that bacterial cytoskeleton-mediated plasmid segregation may utilize a chemical gradient mechanisms rather than polymer ratchet mechanisms e.g. Vecchiarelli et al, 2014, PNAS, 111:4880.

Reviewer #2:

This paper reports analysis of polymerization dynamics of a tubulin-like protein from a bacteriophage using TIRF microscopy of pure protein and *in vivo* imaging. The conclusions of the paper are overall exciting, in that the authors report dynamic instability of individual filaments, and a structure resembling a bipolar mitotic spindle in infected cells. Dynamic instability of a bacterial polymer is not new, but the spindle-like organization is, and it’s very interesting. Thus, the paper may be suited for *eLife*. That said, the paper tries to cover a lot of ground, and the most important conclusions are not sufficiently supported by the data provided. In particular, important claims made on the basis of single filament tracking in living cells seem to be serious over-interpretations because the images provided do not convincingly demonstrate that single filament tracking is possible in living cell. The large degree of over-interpretation of the *in vivo* imaging data that support the most interesting aspects of the paper's conclusion make the current paper unsuited for publication.

Specific points:

1) The most interesting aspect of the paper is the claim of spindle-like organization in infected cells, and dynamic instability of polarized filaments selectively at the center of the cell. This part of the paper is not adequately supported by the data provided. The individual filament traces in Figure 3 are unconvincing, with lots of datapoints consisting of a single point that goes up or down compared to a trendline. This could be noise. The images in Figure 2 and visual inspection of the movies provided suggests tracking of individual filament ends may not be possible using the current microscopy methods. The tracking data and movies appear broadly consistent with the hypothesis provided by the authors, but given the difficulty of convincing single filament tracking, these data have been seriously over-interpreted in the text.

Another strong statement in the text that is not adequately supported by the data is that the filaments are arranged in a bipolar manner, with all fluctuating ends oriented towards the center of the cell. Again, the movies are visually consistent with such an interpretation, but they fall well short of proving it, as implied in the text. Filaments in infected cells appear tightly bundled near the poles, which would make is difficult to detect fluctuations in length of individual filaments there. The imaging in the center of the cell is too imprecise to convincingly detect non-fluctuating ends, or even convincingly track fluctuating ends. The authors’ model or polarized organization is attractive, but they seriously over-interpret the data in trying to support it.

2) The pure filament studies convincingly demonstrate both highly polarized net assembly, leading to treadmilling of uncapped filaments, and dynamic instability of free ends of the kind that prefer to grow; let's call them plus ends. This is very nice data. However, in interpreting the *in vivo* imaging, the authors seem to ignore the possibility of minus ends shrinking. In uninfected cells, shouldn't we see some minus end shrinkage? In infected cells, their cartoon model puts in some kind of nucleating site/cap at poles that orients filaments and caps minus ends. But there is no strong support for this nucleating site from the *in vivo* imaging data. The text states that filaments are randomly oriented in uninfected cells, and bipolar in infected, but this large difference in organization is not well supported by the data provided.

Points 1) and 2) could perhaps be addressed by some kind of quantitative image analysis that focused less on single filament tracking, which is frankly unconvincing, and more on other kinds of fluctuation analysis. For example, a spatial map of temporal variance might demonstrate in a convincing manner that fluctuations are located at random positions in uninfected cells, and concentrated at the center in infected cells. Difference imaging; subtracting frame n- from frame n- might also be beneficial in analyzing where in the cell polymerization dynamics occur. Difference imaging sometimes allows detection of fluctuations even in densely packed regions like the poles. A spatial map of average signal in difference images, which is related to an analysis of temporal variance, could be useful to map the positions in a cell where length change is occurring. More critical interpretation of imaging data is also essential. The authors must distinguish much more precisely between experimental observation and model. It's fine to propose a model that is not completely proven, but is consistent with the data. It is not ok to make statements about what has been observed when the data do not support those statements. It may be the case that the authors will have to argue their model without use of single filament tracking *in vivo*, if such tracking is simply unconvincing or impossible, as suggested by the current Movies and Figures.

3) Figure 6 is not very relevant to the main point of the paper, and the result could be mentioned in words only. In general, the paper jumps around a lot, and would benefit from more thorough analysis of polymerization dynamics, *both in vitro* and (especially) *in vivo*, and less distraction from analysis of DNA.

---

## [Author Response]

Reviewer #1:

*1) It would be useful to present a small summary table comparing the four parameters of dynamic instability (i.e. growth rate, shrinkage rate, catastrophe frequency and rescue frequency) displayed by the phage tubulin compared to eukaryotic tubulins*.

This has been added as Table 1 and references to the rates and table added to the text.

*2)*
Figure 7
*shows a drawing that depicts some type of PhuZ organizing center that anchors the minus ends at the poles of the cell. What is the evidence for this, and are there any molecular candidates for this function (in the Shapiro lab model for the ParA spindle, TipN fulfills this function)?*

We hypothesize the existence of a polar anchoring protein encoded on the phage genome due to the observation that only phage-infected cells show polar stabilization of filaments and the structural similarities to a bipolar spindle. Uninfected cells expressing GFP-PhuZ alone never exhibit this property. We do not currently know which protein fulfils this role, although we are screening the phage genome to identify candidates. It is unclear if this putative protein specifies polar location *de novo* or recognizes a host protein already localized to the pole. We have clarified this in the Discussion and in the Figure 9 legend.

*3) I think the discussion would be more interesting if the authors noted that other workers have also proposed the existence of spindle-like structures that are responsible for prokaryotic DNA segregation, e.g. by using actin-like filament depolymerase motors to mediate plasmid segregation e.g. Ptacin et al, 2010, Nat Cell Biol. 12:791. Moreover, other workers have proposed that bacterial cytoskeleton-mediated plasmid segregation may utilize a chemical gradient mechanisms rather than polymer ratchet mechanisms e.g. Vecchiarelli et al, 2014, PNAS, 111:4880*.

A paragraph discussing these points has been added to the Discussion.

Reviewer #2:

*This paper reports analysis of polymerization dynamics of a tubulin-like protein from a bacteriophage using TIRF microscopy of pure protein and in vivo imaging. The conclusions of the paper are overall exciting, in that the authors report dynamic instability of individual filaments, and a structure resembling a bipolar mitotic spindle in infected cells. Dynamic instability of a bacterial polymer is not new, but the spindle-like organization is, and it’s very interesting. Thus, the paper may be suited for eLife. That said, the paper tries to cover a lot of ground, and the most important conclusions are not sufficiently supported by the data provided. In particular, important claims made on the basis of single filament tracking in living cells seem to be serious over-interpretations because the images provided do not convincingly demonstrate that single filament tracking is possible in living cell. The large degree of over-interpretation of the in vivo imaging data that support the most interesting aspects of the paper's conclusion make the current paper unsuited for publication*.

As per Reviewer #2’s request, we have renamed the figures and reworded the conclusions so that it is clear when we are proposing a model to explain the results as well as to acknowledge a wider range of possible interpretations of our data.

Specific points:

*1) The most interesting aspect of the paper is the claim of spindle-like organization in infected cells, and dynamic instability of polarized filaments selectively at the center of the cell. This part of the paper is not adequately supported by the data provided. The individual filament traces in*
Figure 3
*G, H are unconvincing, with lots of datapoints consisting of a single point that goes up or down compared to a trendline. This could be noise. The images in*
Figure 2
*and visual inspection of the movies provided suggests tracking of individual filament ends may not be possible using the current microscopy methods. The tracking data and movies appear broadly consistent with the hypothesis provided by the authors, but given the difficulty of convincing single filament tracking, these data have been seriously over-interpreted in the text*.

We agree that the dynamic properties of the filaments were not clearly presented in the original Figures and we thank the reviewer for pointing this out. We have revised the Figures to more clearly show filament dynamics and we have included a mutant, GFPPhuZD190A, that is defective in GTP hydrolysis as a control for noise.

Figure 3 now clearly shows an example of a filament undergoing growth and shrinkage in uninfected cells. A second example showing dynamic instability of filaments within uninfected cells is shown in the kymograph in Figure 3.

The time-lapse sequence in Figure 5 and Video 5 have been modified so that the ends of individual filaments can be clearly tracked over time. By following the arrows that mark the filament ends near the center of the cell, growth and shrinkage can be clearly observed.

Figure 5 show measured filament length changes over time normalized to their starting position and Figure 5 shows the average amount of absolute filament length change that occurs per unit time (0.5 s). Wild type GFP-PhuZ filaments exhibit significant and rapid changes in length over time while a mutant filament that is incapable of depolymerizing displays very little change in length over time, demonstrating that the changes in length observed for wild type are not due to noise but are GTP-dependent changes in filament length. It’s also worth noting that many of the changes in length in Figure 5 are large enough (>0.5 μm) to be beyond the noise of most high-resolution optical systems.

*Another strong statement in the text that is not adequately supported by the data is that the filaments are arranged in a bipolar manner, with all fluctuating ends oriented towards the center of the cell. Again, the movies are visually consistent with such an interpretation, but they fall well short of proving it, as implied in the text. Filaments in infected cells appear tightly bundled near the poles, which would make is difficult to detect fluctuations in length of individual filaments there. The imaging in the center of the cell is too imprecise to convincingly detect non-fluctuating ends, or even convincingly track fluctuating ends*.

As this reviewer has correctly noted, the ends of the filaments near the cell middle frequently appear less intense than near the vertex, suggesting that they represent either individual filaments or at least smaller bundles of filaments relative to the rest of the spindle. Despite the lower intensity of these filaments, we can accurately track their ends as shown in Figure 5 and Video 5. Analysis of centrally located filament ends show that they are dynamic, and fluctuate back and forth as shown in Figure 5, Video 5, and Figure 5, Figure 5. In Figure 5 and in the kymograph in Figure 4, the centrally located filament ends can be observed to grow and shrink over time, while the ends of the filaments nearest the pole remain relatively stationary. For comparison, the kymograph in Figure 4 of GFP-D190APhuZ shows that mutant filaments do not change in length over time. Taken together with the *in vitro* data, the simplest model suggests that the filament ends located toward the middle require GTP to grow and shrink. In the text we have softened our conclusion and point out that this is one possible interpretation of the data and that we cannot rule out the possibility that filament or bundle growth are occurring at the vertex of the spindle.

The authors’ model or polarized organization is attractive, but they seriously over-interpret the data in trying to support it.

We have softened the interpretation to indicate that one model consistent with the data is that there is a dramatic change in filament organization upon phage infection.

*2) The pure filament studies convincingly demonstrate both highly polarized net assembly, leading to treadmilling of uncapped filaments, and dynamic instability of free ends of the kind that prefer to grow; let's call them plus ends. This is very nice data. However, in interpreting the in vivo imaging, the authors seem to ignore the possibility of minus ends shrinking*.

This is a very good point. Filaments within uninfected cells are short, dynamic and tend to depolymerize before reaching more than 1 micron in length. As now shown in the time-lapse sequence in Figure 3 and in the kymograph in Figure 3, filaments elongate and depolymerize and we have no way to accurately distinguish plus from minus ends. We assume, of course, that there is some depolymerization from the minus ends, but since we do not know which end is which, we have refrained from speculating directly on this point. Our conclusions from the *in vivo* imaging data of uninfected cells are that filaments display dynamic instability and generally recapitulate the behavior of filaments assembled *in vitro*.

*In uninfected cells, shouldn't we see some minus end shrinkage? In infected cells, their cartoon model puts in some kind of nucleating site/cap at poles that orients filaments and caps minus ends. But there is no strong support for this nucleating site from the in vivo imaging data*.

In the model figure we hypothesize the existence of a polar anchoring protein encoded on the phage genome due to the observation that only phage-infected cells show polar stabilization of filaments and the structural similarities to a bipolar spindle. Uninfected cells expressing GFP-PhuZ alone never exhibit this property. We do not currently know which protein fulfils this role. We have clarified this in the Discussion and the Figure 9 figure legend.

*The text states that filaments are randomly oriented in uninfected cells, and bipolar in infected, but this large difference in organization is not well supported by the data provided*.

In order to underscore the differences in the organization of polymers between uninfected and infected cells, we have added several Figure panels and clarifying text.

Figure 3 show the distribution and dynamics of filaments in a population of uninfected cells. Filaments can now clearly be seen to be different from filaments assembled during phage infection as shown in Figures 4 and 5. We have also clarified our hypothesis and reasoning for the inclusion of a hypothetical “stabilizing” protein in our model. In infected cells, the wishbone-shaped bundles seem very stably anchored at the poles. While these are presumably assembled from many underlying filaments whose individual dynamics are unresolvable, the net effect is a stable (let’s call it) minus end. We presume that the filaments within uninfected cells are randomly oriented within the cell due to the fact short (∼0.2μm) spontaneously assembled filaments can be observed to “tumble” or rotate 360 degrees in the cell during time-lapse microscopy experiments.

*Points 1) and 2) could perhaps be addressed by some kind of quantitative image analysis that focused less on single filament tracking, which is frankly unconvincing, and more on other kinds of fluctuation analysis. For example, a spatial map of temporal variance might demonstrate in a convincing manner that fluctuations are located at random positions in uninfected cells, and concentrated at the center in infected cells. Difference imaging; subtracting frame n- from frame n- might also be beneficial in analyzing where in the cell polymerization dynamics occur. Difference imaging sometimes allows detection of fluctuations even in densely packed regions like the poles. A spatial map of average signal in difference images, which is related to an analysis of temporal variance, could be useful to map the positions in a cell where length change is occurring. More critical interpretation of imaging data is also essential. The authors must distinguish much more precisely between experimental observation and model. It's fine to propose a model that is not completely proven, but is consistent with the data. It is not ok to make statements about what has been observed when the data do not support those statements. It may be the case that the authors will have to argue their model without use of single filament tracking in vivo, if such tracking is simply unconvincing or impossible, as suggested by the current Movies and Figures*.

As noted above, we have significantly revised the figures so that changes in filament lengths can be clearly seen for both infected and uninfected cells. We acknowledge that our model is not completely proven; there is certainly more work to do, however; together with the Movies and the *in vitro* data, we suggest that our *in vivo* data are fully consistent with and supportive of our model. We have reworded some of our conclusions about the data to avoid confusion and convey a better sense of the unresolved issues. We agree with the reviewer that difference mapping is a powerful tool that can often eliminate some of the perils of tracking individual filaments, but the limitations of our host/phage system have complicated its use in this case. Filaments of PhuZ in infected cells are always a mixed population of labeled (from a plasmid) and unlabeled (from the phage) monomers; movement and/or exchange of the two populations of monomers within a polymer, as well as movement of the polymer itself in all three axes, leads to variations in intensity of the pixels and leads to a “maxed out” standard deviation for the entire filament. We believe that traditional kymographs and tracking individual filament ends as shown in the revised figures allow for a better evaluation of the change over time for each of the filament ends.

*3)*
Figure 6
*is not very relevant to the main point of the paper, and the result could be mentioned in words only. In general, the paper jumps around a lot, and would benefit from more thorough analysis of polymerization dynamics, both in vitro and (especially) in vivo, and less distraction from analysis of DNA*.

We understand and acknowledge Reviewer #2’s concern. These data are significant because they begin to address the purpose of the centering function and how it is coordinated with other essential processes of the phage *in vivo*. Is DNA replication required for the spindle to assemble or to function properly? We show that it is clearly not required and this eliminates an entire set of models and possible lines of inquiry. For these reasons, we have retained this data set in the revised manuscript. We have clarified these points in the text.